

# Occurrence and abundance of microplastics in surface water of Songkhla Lagoon

Siriporn Pradit[1,2], Prakrit Noppradit[1,2], Kittiwara Sornplang[1,2], Preyanuch Jitkaew[1,2], Thanakorn Jiwarungrueangkul[2,3] and Dudsadee Muenhor[4,5,6]

[1] Marine and Coastal Resources Institute, Faculty of Environmental Management, Songkhla, Thailand
[2] Coastal Oceanography and Climate Change Research Center, Faculty of Environmental Management, Prince of Songkla University, Songkhla, Thailand
[3] Marine Environment and Geoinformatics Technology Research Unit, Faculty of Technology and Environment, Prince of Songkla University, Phuket, Thailand
[4] Faculty of Environmental Management, Prince of Songkla University, Hat Yai, Songkhla, Thailand
[5] Health Impact Assessment Research Center, Prince of Songkla University, Hat Yai, Songkhla, Thailand
[6] Center of Excellence on Hazardous Substance Management (HSM), Bangkok, Thailand

Corresponding author
Dudsadee Muenhor,
dudsadee.m@psu.ac.th

## ABSTRACT

**Background:** Microplastic (MP) pollution is now a global critical issue and has been the subject of considerable worry for multiple various types of habitats, notably in lagoons which are coastal areas connected to the ocean. MPs are of concern, particularly because floating MP in surface water can be ingested by a number of marine organisms. There are several lagoons along Southeast Asia's coastline, but Songkhla Lagoon is Thailand's only exit with a rich biodiversity. To date, there has been little research undertaken on MP in this lagoon, so there is a pressing need to learn more about the presence of MP in the lagoon's water.

**Methods:** We investigate MPs in the surface water of Songkhla Lagoon, Thailand. Sampling took place at ten stations in the lagoon during the wet season in December 2022 and the dry season in February 2023. Samples were digested with hydrogen peroxide to remove organic matter followed by density separation using saturated sodium chloride. MPs were visually examined under a stereo microscope to describe and determine the shape, size, and color. Polymer type was identified using a micro Fourier transform infrared (FTIR) spectrometer. Moreover, the *in-situ* of water quality of the surface water was measured using a multi-parameter probe. A Mann-Whitney U test was performed to investigate the variations in MP levels and water quality parameters between the wet and dry seasons. Correlation analysis (Spearman rho) was used to determine the significance of correlations between MP and water quality ($p < 0.05$).

**Results:** MPs were detected at all ten of the sites sampled. The most abundant MPs were small size class (<500 μm, primarily consisting of fibers). Five types of polymers were seen in surface water, including polyethylene terephthalate, rayon, polypropylene, polyester, and poly (ethylene:propylene). Rayon and polyester were the dominant polymers. Additionally, the most dominant color of MPs in the wet and dry season was black and blue, respectively. The mean contents of MPs in the wet

and dry season were $0.43 \pm 0.18$ and $0.34 \pm 0.08$ items/L, respectively. The Mann-Whitney U test suggested a significant difference between water quality in the wet and dry seasons ($p < 0.05$). Correlation analysis (Spearman rho) indicated a negative significant difference relationship between the MPs and the values of total dissolved solid (TDS) in the wet season ($r = -0.821$, $p = <0.05$), revealing that the large amounts of MPs may possibly be dispersed within surface water bodies with low TDS concentrations. Based on the overall findings, MP pollution in the surface water of the lagoon is not found to be influenced by the seasonal context. Rivers flowing into the lagoon, especially the U-Taphao River, may be a principal pathway contributing to increased MP pollution loading in the lagoon. The results can be used as baseline data to undertake further research work relevant to sources, fates, distribution, and impacts of MPs in other coastal lagoons.

# INTRODUCTION

Plastics are now extensively produced and applied in industrial and consumer sectors with a global production of 390.7 million metric tons in 2021 (*Statista, 2023*). Improperly handled waste plastics end up in the ocean, for instance in 2010, 192 coastal countries created 275 million metric tons of plastic waste, of which about 4.8–12.7 million metric tons were lost to the ocean (*Jambeck et al., 2015*). As a result of degrading processes, plastic waste fragments into a variety of sizes (*Hidalgo-Ruz et al., 2012*) and enters the marine environment with varying chemical composition, specific density, color, and form (*Duis & Coors, 2016*). A tiny size called microplastic (MP) is generally defined as small particles with a diameter between 1 μm to 5 mm (*Cole et al., 2011*; *Thompson et al., 2009*). Two categories of MPs have broadly contaminated the global ocean: primary and secondary MPs (*Boucher & Friot, 2017*). MPs are categorized as primary when intentionally manufactured at micro sizes for specific purposes and secondary if fragmented from bigger plastic objects (*Lehtiniemi et al., 2018*) or when they undergo degradation by various environmental factors over a long period of time (*An et al., 2020*). MPs in seawater were among the first to be discovered during the literature search, producing the most search results (*Kye et al., 2023*). MPs occur in various oceans, for example: the South China Sea (*Tan et al., 2020*), Antarctic Ocean (*Cincinelli et al., 2017*), and Midwest Pacific (*Wang, Ge & Yu, 2020*). Additionally, around 15–51 trillion MPs float on the ocean's surface (*van Sebille et al., 2015*) and secondary MPs are the majority of MPs observed in the ocean (*Rochman et al., 2019*).

MPs are of concern, particularly because they can be ingested by a number of marine organisms (*Pradit et al., 2020*). Misidentification of MPs during feeding can lead to direct ingestion (*Ory et al., 2017*). MPs can absorb additional chemicals (plastic additives) including persistent organic pollutants (POPs), and heavy metals contained in seawater

(*Goh et al., 2022*; *Jualaong et al., 2021*). As a consequence, there are numerous negative effects, including mortality, decreased feeding activity, inhibition of growth and immunity, and the development of endocrine disruption, oxidative stress, neurotransmission failure, and even genotoxicity (*Lu et al., 2016*). However, the concern of MP contamination in the marine environments is of emerging critical global concern due to their ubiquitous occurrence across all ocean watersheds, habitats, ecosystems, food chains, and food webs, along with health effects on both humans and other living organisms, which has prompted the emergence of problem-solving measures (*Huang et al., 2021*; *Rochman, 2020*). Moreover, there is growing scientific evidence indicating that marine organisms which are finally consumed by humans may themselves feed on MPs directly from seawater or from lower trophic levels (*Tanaka & Takada, 2016*).

MPs are more concentrated nearshore or in estuaries near land than in open seas (*Zhang et al., 2020*). They are also more concentrated in seas with physical characteristics such as semi-enclosed bays than on open seashores (*Zhu et al., 2019*). Lagoons are in transitionary zones between land and sea, and they are strongly subjected to human impacts and natural constraints. Consequently, these coastal ecosystems are very dynamic and highly productive (*Silva et al., 2013*). Anthropogenic forces can also exacerbate MPs and chemical and microbial contamination in lagoons. Songkhla Lagoon is the biggest lagoonal water body in Thailand and Southeast Asia (*Ganasut, Weesakul & Vongvisessomjai, 2005*). It is extremely rich in biodiversity and provides resources that are essential to the wellbeing, livelihoods, and health of local adjacent communities including fish for food and markets and as a water source. Additionally, the livelihood of people living in twenty-five districts of Songkhla, Phatthalung, and Nakhon Si Thammarat Provinces rely on the lagoon's fishery resources (*Hue et al., 2018*). Songkhla lagoon is a lagoon in southern Thailand which is made up of four interconnected sections. At present, there is an increase in population, economic development, and infrastructure development in the surrounding areas. A possible driver of the plastic/MPs pollution that can be stressor on the lagoon include the lack of garbage management in some areas, sewers from human activity, washing activity, and fishing activity (*Pradit et al., 2021*). In addition, rubber, parawood, and seafood processing industries are the major sources of garbage dumped into the U-Taphao River at a rate of 41,000 m$^3$/day (*Sirinawin & Sompongchaiyakul, 2005*). Moreover, the lower section (southernmost of the lagoon) is facing a significant decline in water quality (*Pornpinatepong et al., 2016*).

To the best of our knowledge, there are few investigations of MPs in Southeast Asia lagoon water, particularly in Thailand, indicating a research gap. Therefore, there is an urgent need to understand how much MPs are in the lagoon's water. The objective of this study is to investigate the occurrence of MPs and the water quality to gain the current condition of the Lagoon's water. Hence, researchers have raised concerns about the lack of guidelines to prevent and mitigate MP contamination in different environments. Therefore, monitoring MPs in the water column necessary to assess the risk that MPs pose to aquatic animals.

## MATERIALS AND METHODS

### Study area

Songkhla Lagoon is a shallow and brackish water lagoon located in Southern Thailand (7° 08′N to 7°50′N and 100°07′E to 100°37′E). The lagoon is connected to the Gulf of Thailand (GoT) *via* an inlet. The lagoon system is divided into four sections. The study was conducted in the lower section which connects to the mouth of the lagoon. The salinity varied from fresh water in the upper section to saltwater in the lower sections. The lower section had an average depth of 1.5 m. Since the only inlet that connects to the sea is quite narrow (about 450 m). The average tidal range at the mouth of the lagoon is approximately 0.5 m (*Lheknim & Yolanda, 2020*), it is classified as a micro tidal estuarine lagoon system. Several rivers and streams flow into the lower section of lagoon, such as: (a) U-Taphao river, the lagoon's major supply of freshwater, which is approximately 120 km long and passes through Hat Yai Municipality before entering; (b) Phawong river which passes through a dense mangrove swamp; and (c) Samrong Canal which is a short canal around 4 km long, passing through Muang Songkhla District before entering the lagoon. The lagoon provides ecosystem services to the region, for example, a high diversity of fish species. Most local people are involved in small-scale fishing in the area.

### Sample collection

Sampling took place at ten stations in the lower section of Songkhla Lagoon (Fig. 1) during the wet season in December 2022 and the dry season in February 2023. Seven of the ten stations were positioned near the river mouth (stations 1, 2, 3, 4, 6, 7, and 10), one near a mangrove (station 5), one near an island inside the lagoon (station 8), and another in the center of the lagoon (station 9). In order to determine MPs present in the water, 50 L of surface water (per station) was collected using a steel bucket (5 L) and filtered it through a 20 µm net. MP samples were then transferred to a glass bottle and cooled to 4 °C for further analysis. Prior to MP collection, an *in situ* water quality measurement was made with a multi-parameter probe (EUTECH, PCD650). Discrete sampling devices, such as the Niskin bottle (*Bagaev, Khatmullina & Chubarenko, 2018*), the rosette (*Dai et al., 2018*), buckets, and bottles (*Khalik et al., 2018*; *Zhu et al., 2019*) were used to collect water samples at precise depths. Because the sampling area in the lagoon is quite shallow and this study only focuses on surface water, using a bucket was deemed appropriate for this investigation. Moreover, the volume of collected water in this study was comparable to that of *Amin et al. (2020)*.

### Microplastic separation and identification

The samples were digested with 20 ml of hydrogen peroxide ($H_2O_2$) (30%) to separate organic matter from MP (*Miller, Kroon & Motti, 2017*). This was followed by density separation to remove MPs from the rest of the sample using a saturated sodium chloride (NaCl) solution (1.2 g/cm$^3$). According to *Galgani et al. (2013)*, NaCl solutions are the most commonly used. After that, MP samples underwent a filtration process, through Whatman glass microfiber fitter, GF/C (47 mm in diameter, 1.2 µm in pore size). After
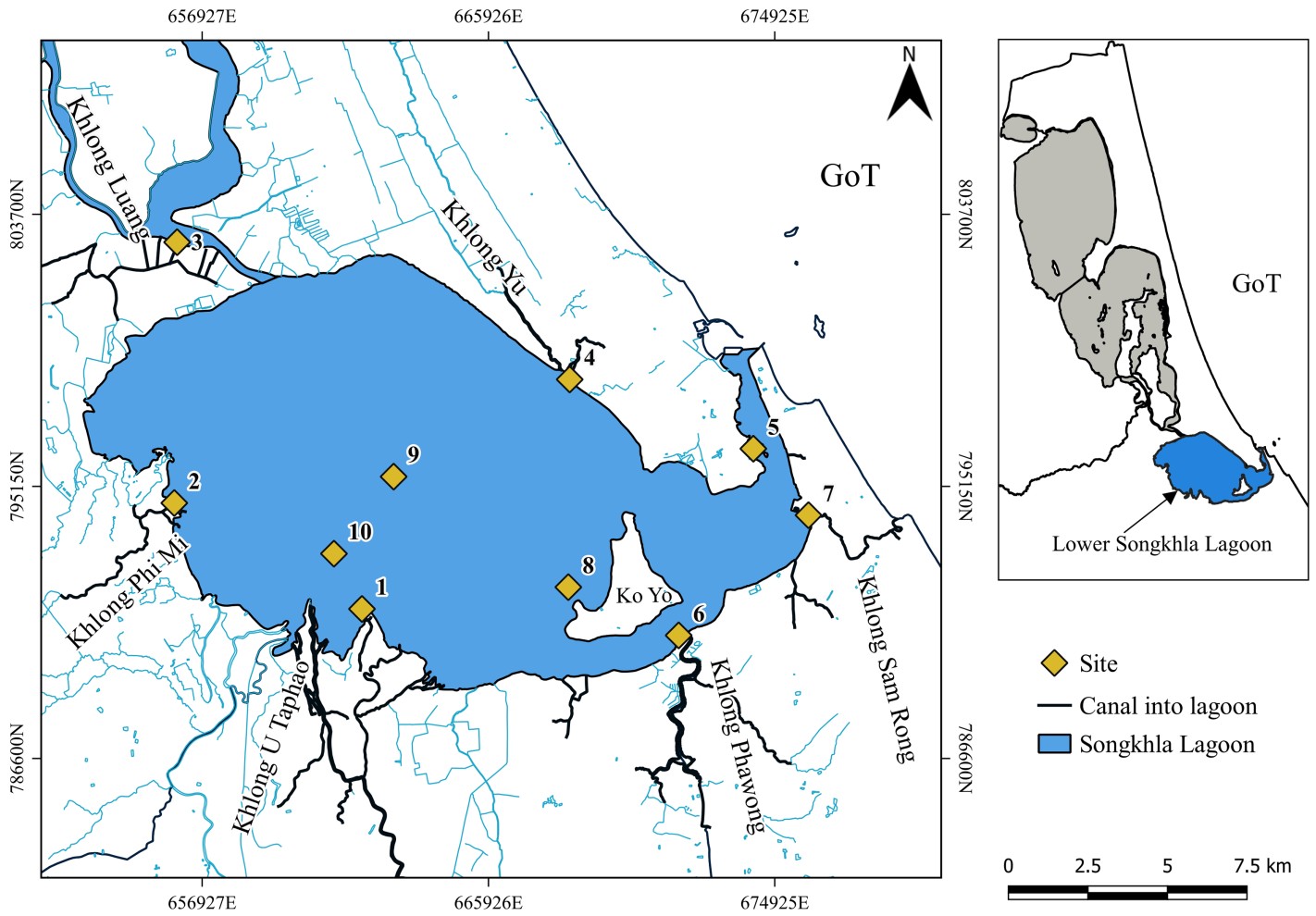

**Figure 1 Map of sampling sites in the lower Songkhla Lagoon.** Source and Printed by GEO-Informatics Center for Natural Resources and Environment, Prince of Songkla University, Thailand.

filtration, the filters were placed in glass Petri dishes and oven-dried at 50 °C for 24 h for further analysis.

All MPs that remained on the glass fiber papers were visually examined under a stereo microscope (Leica EZ4W; Leica system, made in Singapore). The characteristics of the MPs were observed visually (*Hidalgo-Ruz et al., 2012*) to describe the shape of particles into fiber and fragment (*Li et al., 2016*), length was classified into three size classes *i.e.*, <500, 500–1,000 and >1,000 μm, as well as colors in four categories (black, blue, transparent, and other). To distinguish between organic and non-organic materials (plastic), we followed *Hidalgo-Ruz et al. (2012)* guidelines. In situations where it was unclear whether the material was plastic or not, we additionally employed the hot needle test (*De Witte et al., 2014*). About 25% of the MPs were selected at random from all samples to confirm their polymer type using a micro Fourier transform infrared spectrometer system (μFT-IR; Spotlight 200i FT-IR microscopy system; PerkinElmer, Waltham, MA, USA) with the attenuated total reflection mode. The wavelengths employed were 4,000 to 400 cm$^{-1}$, with a resolution of 4 cm$^{-1}$. The sample's spectrum was compared

to known polymers database (PerkinElmer Polymer database) and the type of plastic was determined when the research score was higher than 80%.

## Contamination prevention

The utilization of plastic items in the sample collection and analysis was avoided whenever possible. For example, metal tweezers and sieves along with glass measuring cylinders, beakers, volumetric flasks. Distilled water and saturated sodium chloride solutions were filtrated through a glass fiber membrane prior to use. Aluminum foil was used to cover the samples and glassware, and the experiments were performed under a fume hood. A blank test was conducted in a laboratory by placing Petri dishes containing distilled water in a laboratory. At the end of the experiment, no MP was found in the Petri dish.

## Data analysis

Statistical analysis was applied to calculate the mean and standard error. Mann-Whitney U test was used to elucidate any differences in levels of MPs and water quality parameters between the wet and dry seasons. Correlation analysis (Spearman's rho) was employed to examine the significance of the relationships between MP concentrations and water quality parameters.

Throughout the present study, probability values of less than 0.05 ($p < 0.05$) were considered as statistically significant.

## RESULTS

### MP abundance in surface water

MP abundance in water samples ranged from 0.25–0.90 (average 0.43 ± 0.06 items/L) and 0.22–0.47 (average 0.34 ± 0.03) in the wet and dry seasons, respectively. The wet season had more MPs than the dry season. In the wet season, MP levels were high (more than 0.30 items/L) particularly in the mouth of the river sites (6 stations: 1, 2, 3, 4, 6, and 10) and at the mangrove site (station 5). During the dry season, MPs were highest at the river mouth (3 stations: 1, 2, and 10), followed by the middle of the lagoon site (station 9) and the mangrove site (station 5). Interestingly, the number of MPs (0.29 items/L) at the site near the island (station 8) remained similar throughout both seasons. MP concentrations in both season are shown in Fig. 2.

### MP characteristics (shape, size, and color)

This study classified MPs into two shapes based on their basic structure: fibers and fragment. Throughout the study, fiber shapes were more common than fragments in both wet (fiber = 83.41%) and dry seasons (fiber = 62.57%). In both seasons, the size class <500 μm was the most common (wet season = 40.55%, dry season = 63.16%), followed by 500–1,000 μm size class (wet season = 38.71%, dry season = 25.15%). Size classes >1,000 μm were seen in small proportions (wet season = 20.74%, dry season = 11.70%). The main colors of the MPs were blue, black, and transparent. The percent of color found in the dry season was; blue>other>black>transparent whereas in wet season it was;

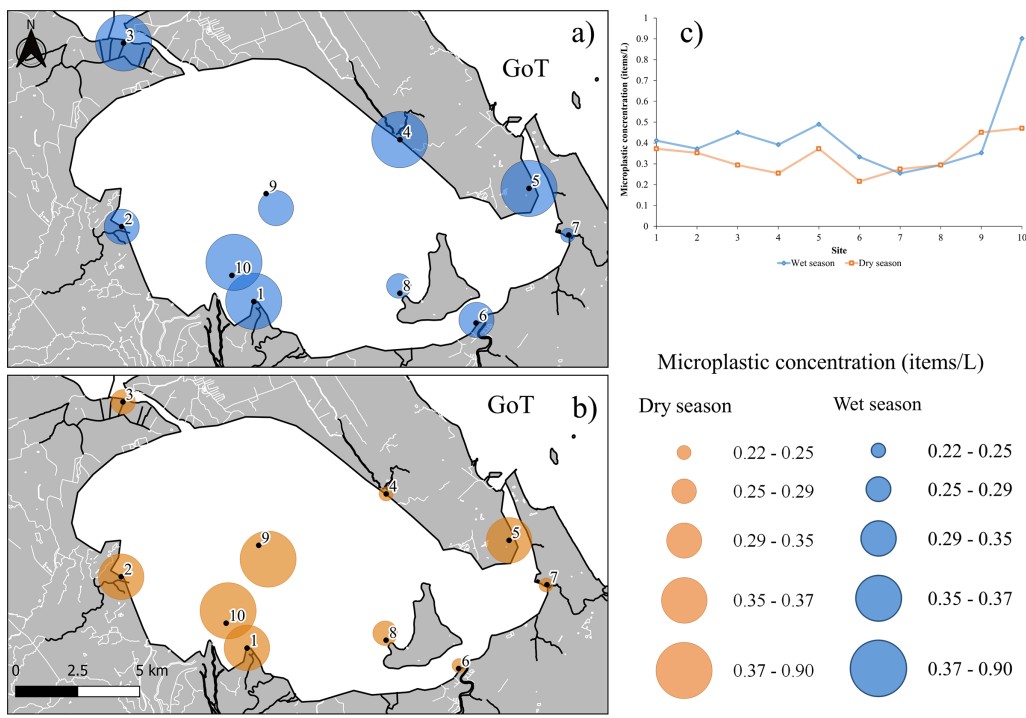

**Figure 2** MP concentrations in Songkhla Lagoon: (A) wet season (blue); (B) dry season (orange); and (C) combination of the two seasons. Source and Printed by GEO-Informatics Center for Natural Resources and Environment, Prince of Songkla University, Thailand.

black>blue>transparent>other. The other colors included red, yellow, green, purple, and pink. Black and blue were the most abundant colors of MPs during both the wet and dry seasons (Fig. 3).

## Polymer identification of MPs

In this study, five types of polymers were seen in surface water (Figs. 4 and 5), including polyethylene terephthalate (PET) (7.14% in wet season and 33.33% in dry season), rayon (35.71% in the wet season and 38.89% in the dry season), polypropylene (PP) (14.29% in the wet season and 16.67% in the dry season), polyester (35.71% in the wet season and 5.56% in the dry season), and poly (ethylene:propylene) (7.14% in the wet season and 5.56% in the dry season). In the wet season, rayon and polyester were the most commonly found, while rayon and PET were the most common in the dry season. Of interest is that rayon was found the most in both seasons with the highest frequency at a proportion of 38.89% in the dry season. Interestingly, based on the FTIR analysis, di-(2-ethylhexyl) phthalate (DEHP), and diethylene glycol (DEG) were detected in water samples in the wet season. DEHP is a plasticizer used to make polymers or plastics flexible (*Kim et al., 2003*), but DEG, an intermediate in polymer production, cannot be identified as a polymer (*Lotti et al., 2006*) (Fig. 6).

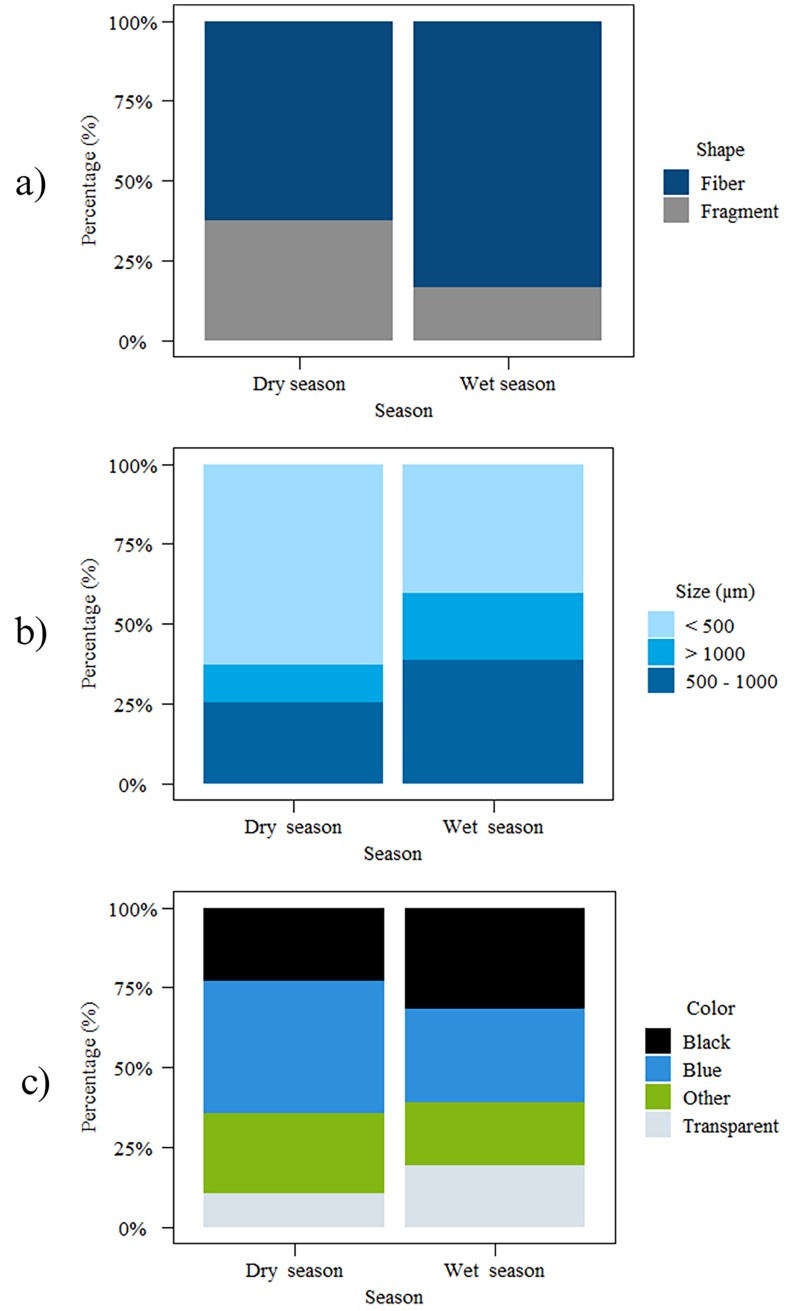

**Figure 3 Characteristics of MPs in Songkhla Lagoon from both wet and dry seasons: (A) shape; (B) size; and (C) color.**

## Water quality of the lagoon

The water quality of the surface water in this study was assessed based on temperature, pH, salinity, conductivity, transparency, total dissolved solid (TDS), and dissolved oxygen (DO), which are presented in Figs. 7–9. Salinity and pH levels in the wet and dry seasons ranged from 0 to 1 ppt and 6.65–8.12 ppt; 6–29 ppt and 6.79–8.24 ppt, respectively. Transparency was determined to be 30–60 cm in the wet season and 30–200 cm in the dry

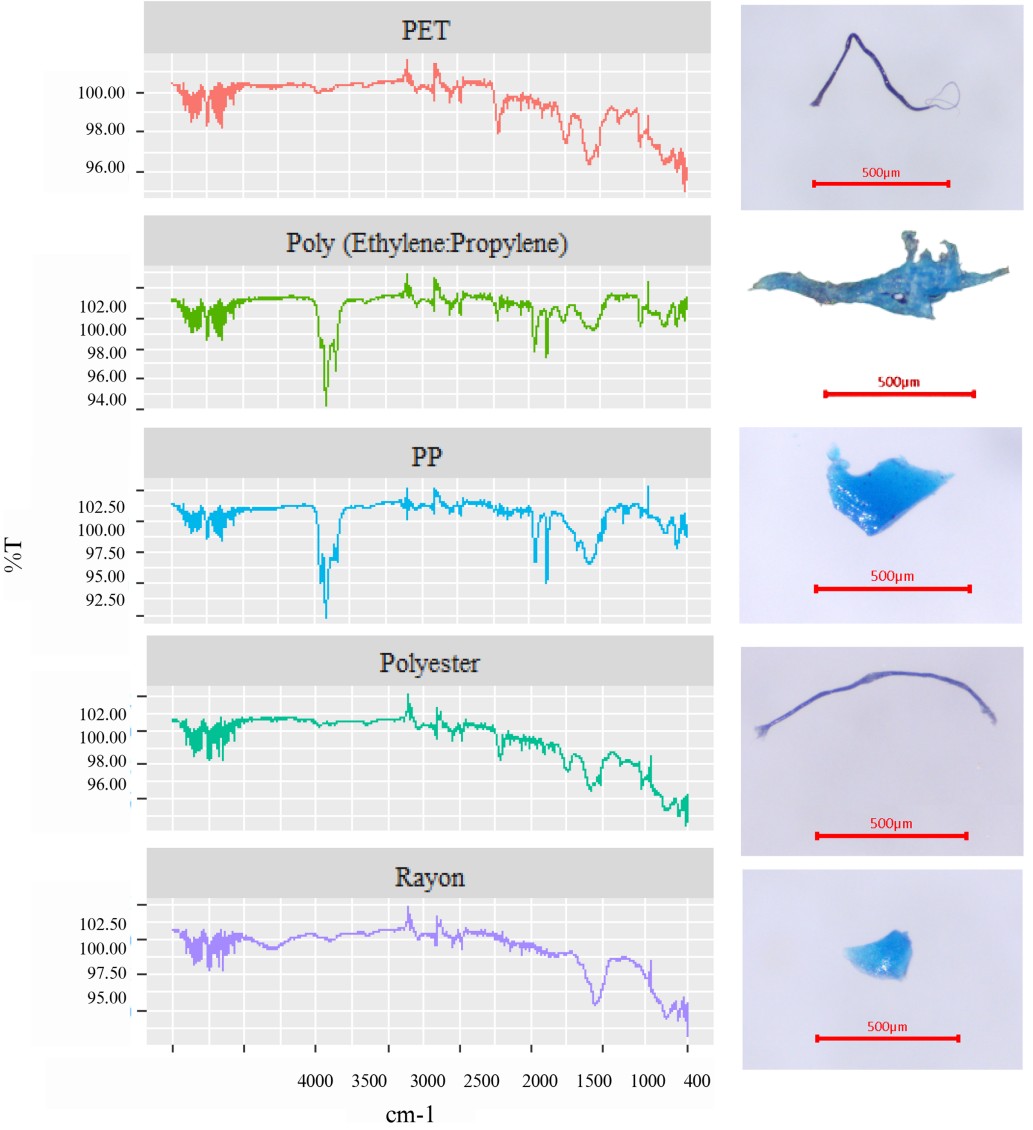

**Figure 4 Five types of polymers found in Songkhla Lagoon in both wet and dry seasons.**

season. DO levels were lowest (0 mg/l in the dry season and 0.22 mg/l in the wet season) at the mouth of the Somrong canal (station 7). TDS levels ranged from 0 to 0.4 ppt during the wet season and 10 to 44 ppt during the dry season. The study shows that the electrical conductivity of the lagoon varied between 0.06–2.27 mS/cm in the wet season and 10.74–46.32 mS/cm in the dry season. Overall, the values of these water quality measures were greater in the dry season compared to the wet season.

## DISCUSSION

### Abundance and distribution of MP

Our findings highlight that high MP concentrations in the mouth area of U-Taphao River (stations 1 and 10) make it a hot spot of MPs in the surface water, followed by the

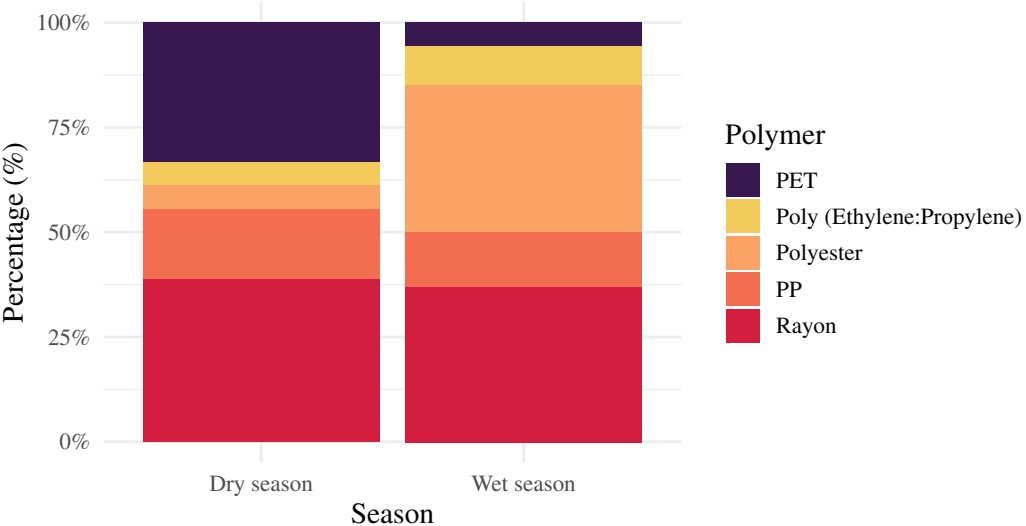

**Figure 5  Polymer types found in wet and dry seasons.**

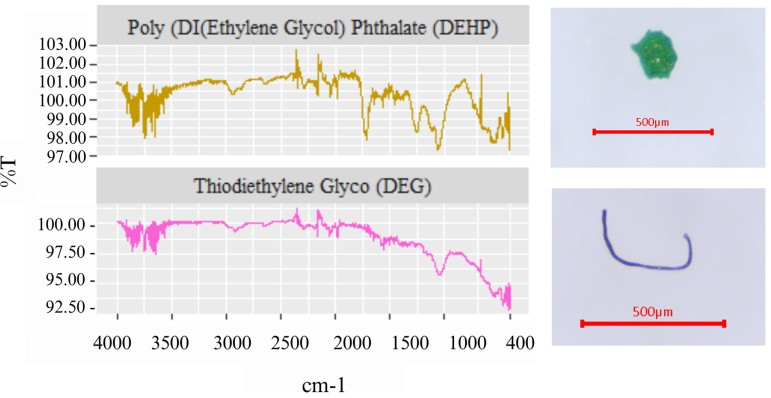

**Figure 6  The spectrum of DEHP and DEG detected in surface water of Songkhla Lagoon.**

mangrove site (station 5) in both seasons. Hence, the U-Taphao River could be an important pathway for the release of MPs to the lagoon, particularly in the wet season. A possible explanation is that the river is long (about 120 km) and flows through densely populated areas as well as many industrial and agricultural areas alongside the river before entering the lagoon compared to the Phawong (station 6) and Samrong (station 7) rivers. It is interesting that the mangrove region (station 5) can be a sink for MPs in surface water because the MP attached with roots. Nonetheless, the mangrove site near the lagoon's mouth can get MP from both seaward and landward sites. MP abundance, distribution, and transport are influenced by a variety of factors, including wind speed, drifting velocity, wave and current speed. Tidal forcing also causes drifter movement with weak winds (*Fitzenreiter, Mao & Xia, 2022*). MP levels were greater in the wet season than in the dry season, which is likely due to heavy rain during the wet season. Since a heavy rainy period in the lagoon is affected by the Northeast monsoon (from October to January), this may

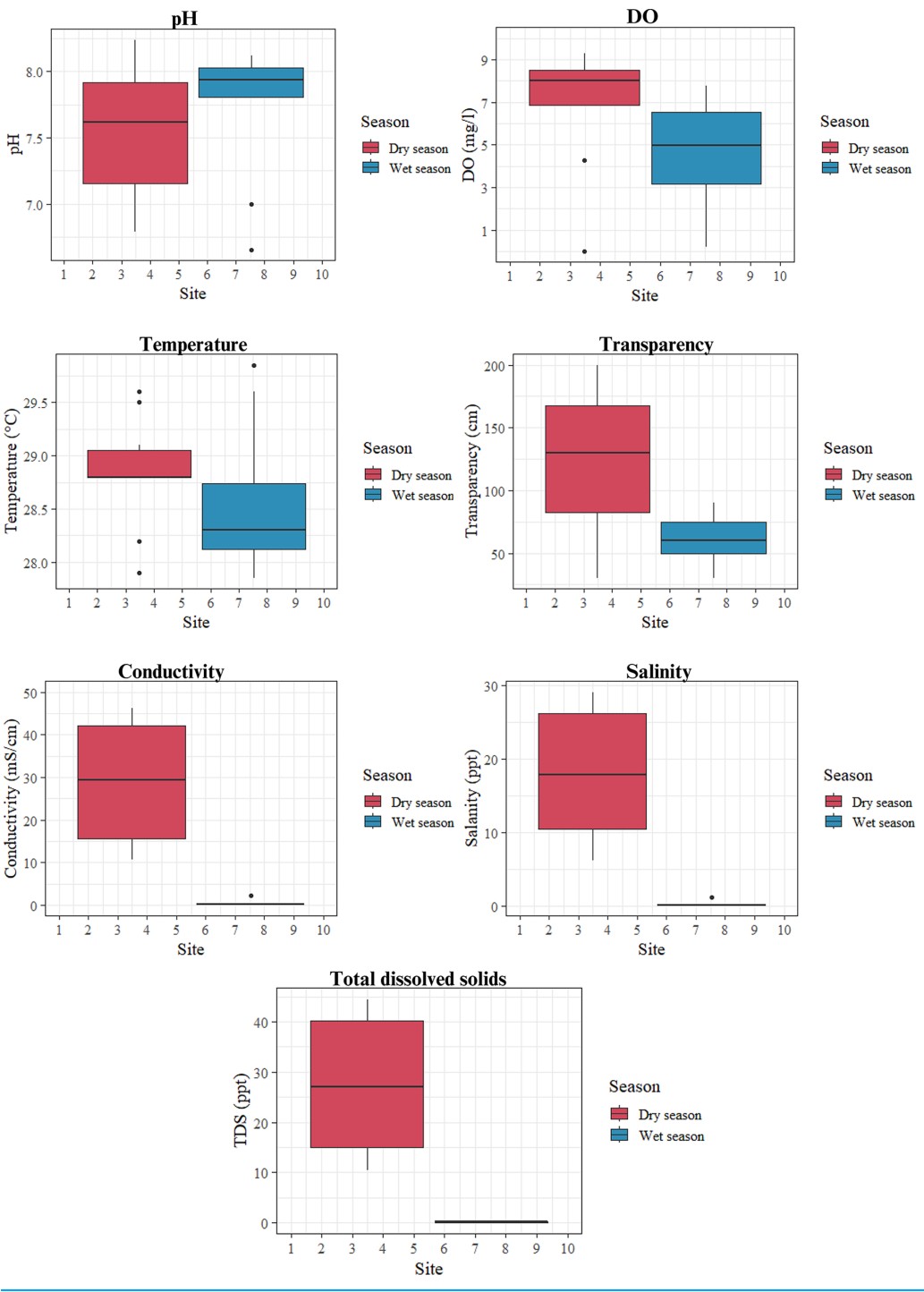

**Figure 7** The pH, DO, temperature, transparency, conductivity, salinity and TDS values of Songkhla Lagoon surface water in both wet and dry seasons.

bring storm water and MPs from the surrounding areas to the lagoon. Due to the weak current in the center of the lagoon (station 9), MPs seem to remain at a similar concentration in this area in both seasons. This can imply that MPs will possibly stay at the same area (in the center) and MPs may move up and down following the tide.

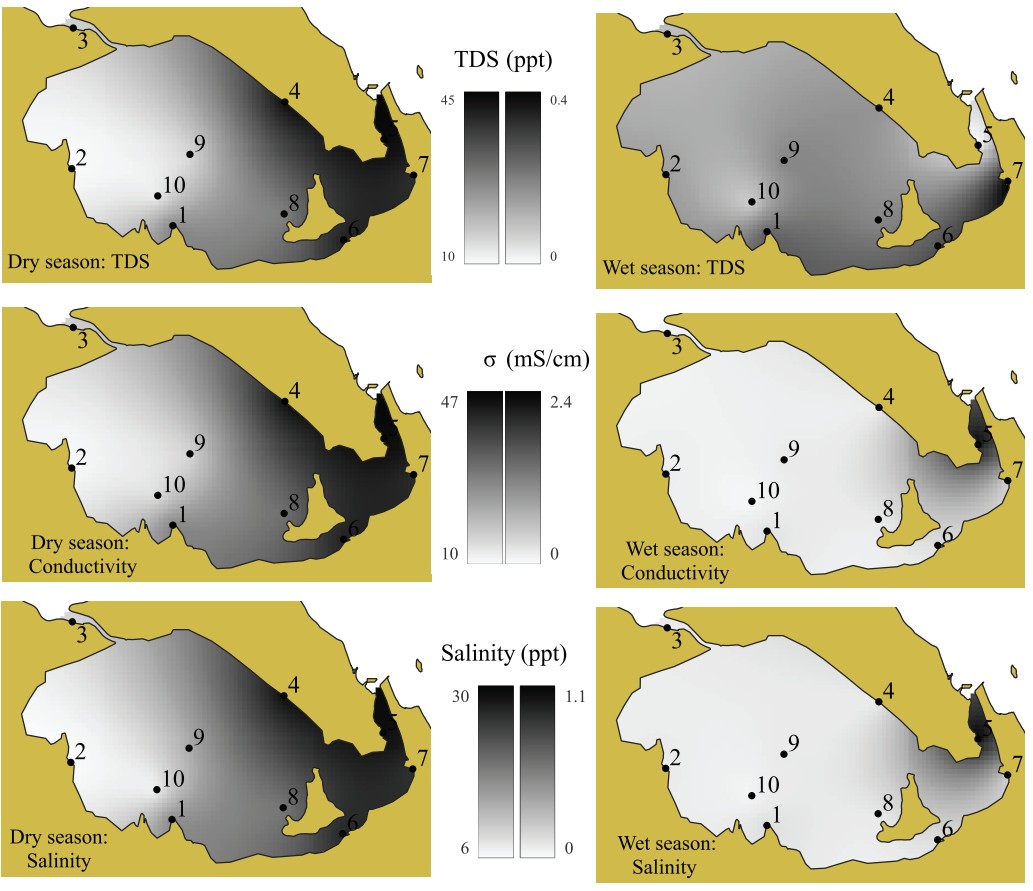

**Figure 8 The distribution of TDS, conductivity and salinity values of Songkhla Lagoon surface water in both wet and dry seasons.** Source and printed by GEO-Informatics Center for Natural Resources and Environment, Prince of Songkla University, Thailand.

Consequently, MPs tend to be consumed more by the fish or shrimp in this area than in locations with well-flowing water. However, this issue is still unclear and further in depth studies are required to understand the drifting behavior of MPs in the lagoon system. Additionally, MPs in the water surface of Songkhla Lagoon could come from households, industry, agriculture, or fishing activities.

The concentration of MP in surface water found in this study was similar to MP in Malaysian wetlands (*Ibrahim et al., 2021*), an estuary in China (*Xu et al., 2018*) and a river in southern Thailand (*Pradit et al., 2023*). Interestingly, several types of polymer type found in China, especially PVC which has a density higher than seawater water, resulting from strong upward and downward movement of water (*Xu et al., 2018*). Never the less our study, the concentration of MPs in water was greater than those detected in the South of Brazil (*e Silva & de Sousa, 2021*), but lower than those observed in Northern Tunisia (*Wakkaf et al., 2020*), Southwest Nigeria (*Olarinmoye et al., 2020*), and an estuary in Malaysia (*Zaki et al., 2021*). Apart from that, the number of MPs was similar to a previous study by *Pradit et al. (2023)* where MPs were investigated in the U-Taphao River which flows into Songkhla Lagoon. A comparison of our results with other aquatic ecosystems is

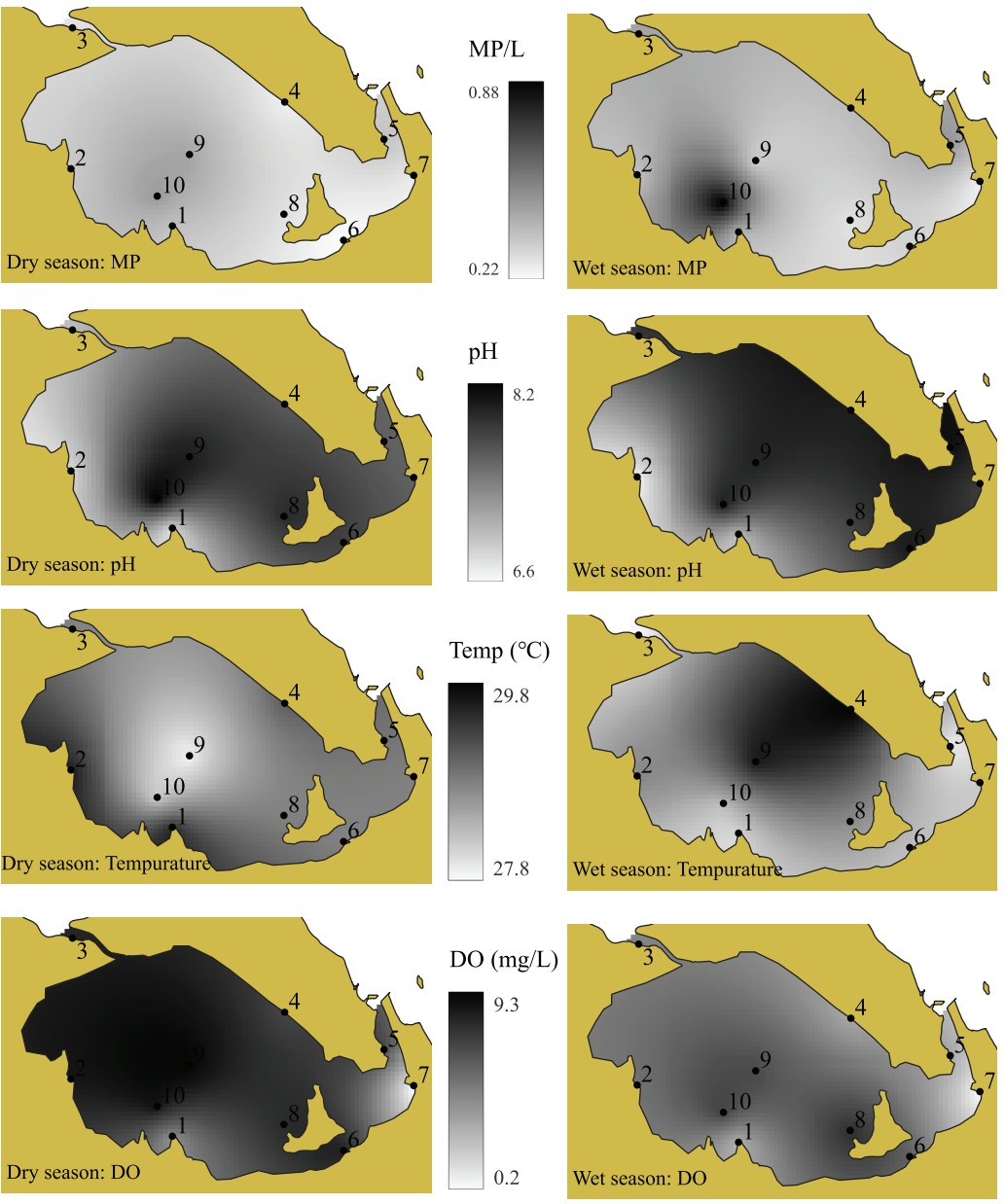

**Figure 9 The distribution of MP, pH, temperature and DO values of Songkhla Lagoon surface water in both wet and dry seasons.** Source and printed by GEO-Informatics Center for Natural Resources and Environment, Prince of Songkla University, Thailand.

summarized in Table 1. Nevertheless, the lower section of the Songkhla lagoons is influenced by tidal action at sea and river flow. Strong variations in physicochemical parameters such as salinity, pH, and temperature exist in the saltwater-freshwater mixing zone. As a result, it may influence the distribution of MPs in this area. However, tropical ecosystems face a rising threat of development and habitat degradation due to population increase and urbanization, agricultural expansion and deforestation, and mining. Some of these activities also cause the discharge of MP or hazardous compounds into the lagoon and coastal areas.

**Table 1 Correlation coefficients between parameters and MP concentrations by wet season and dry season.**

| | pH | Temperature | TDS | Salinity | DO | Conductivity | Transparency | MPs |
|---|---|---|---|---|---|---|---|---|
| **Wet season** | | | | | | | | |
| pH | 1.000 | −0.055 | −0.273 | 0.298 | 0.055 | 0.273 | −0.322 | 0.309 |
| Temperature | | 1.000 | 0.170 | −0.299 | 0.347 | −0.255 | 0.457 | −0.365 |
| TDS | | | 1.000 | 0.450 | −0.261 | 0.455 | −0.118 | −0.806** |
| Salinity | | | | 1.000 | −0.450 | 0.997** | −0.631 | −0.365 |
| DO | | | | | 1.000 | −0.442 | −0.068 | −0.006 |
| Conductivity | | | | | | 1.000 | −0.613 | −0.382 |
| Transparency | | | | | | | 1.000 | 0.297 |
| MPs | | | | | | | | 1.000 |
| **Dry season** | | | | | | | | |
| pH | 1.000 | −0.782** | −0.006 | 0.067 | 0.539 | −0.006 | 0.626 | 0.195 |
| Temperature | | 1.000 | 0.175 | 0.131 | −0.488 | 0.175 | −0.690* | −0.239 |
| TDS | | | 1.000 | 0.988** | −0.636* | 1.000** | −0.359 | −0.549 |
| Salinity | | | | 1.000 | −0.588 | 0.988** | −0.322 | −0.494 |
| DO | | | | | 1.000 | −0.636* | 0.644* | 0.439 |
| Conductivity | | | | | | 1.000 | −0.359 | −0.549 |
| Transparency | | | | | | | 1.000 | 0.440 |
| MPs | | | | | | | | 1.000 |
| N | 10 | 10 | 10 | 10 | 10 | 10 | 10 | 10 |

**Note:**
* Correlation is significant at the 0.05 level (2-tailed).
** Correlation is significant at the 0.01 level (2-tailed).

To create a comprehensive process, field blanks, procedural blanks, and positive or negative controls must be implemented in the study design. Though we did not carry them out in our study, we nonetheless make every effort to avoid environmental contamination throughout both laboratory activities and fieldwork. The additional limitations of the study include the shallowness of the lagoon, which made it necessary to choose the appropriate sampling water with a bucket rather than a manta net. The possibility of overestimation or underestimation may be connected to the sampling time (*e.g.*, low tide or high tide), the volume of water sampled, and the sampling techniques used.

### Shape, size, and color of MPs

Shape: determining the shape of MP is an interesting aspect of MP research because it is one of the best indicators of its origin (*Ugwu, Herrera, & Gómez, 2021*). It was known that fiber is most common in the surface water of rivers, lagoons, and the sea. This is also found to be the case in the present study, with fiber the most predominant type found in the surface water of Songkhla Lagoon. This is comparable to a study of MPs in Malaysian seawater (*Amin et al., 2020*), which reported that the majority of fibers were detected at the near shore station compared to other areas. The fiber shape in the wet season was higher than in the dry season. There is a possibility of breakage from textile materials or ropes

from fishing equipment in the lagoon. The loading of MP from fishing nets can contribute to an increase of fiber shapes in the environment (*Anbumani & Kakkar, 2018*). Beside this, fiber shapes can be from laundry (*Pradit et al., 2021*). *De Falco et al. (2018)* found that textile washing methods discharge about 6 million microfibers of woven polyester each 5 kg load, supporting our observation. Moreover, fiber shaped MPs were more likely to be ingested by marine organisms (*Kain et al., 2016*). *Jitkaew et al. (2024)* report that fiber shapes were more commonly ingested by catfish in the lagoon. However, fragments is one of the most common type found after plastic breaks down by ultraviolet light and mechanical forces from wind and waves (*Caldwell et al., 2019*), but in our study we found less fragments than fiber. However, fragments can be formed from consumer products such as plastic bags, bottles, and containers.

Size: from the breakdown of large pieces of materials, whether they are fibers or fragments, the predominant size detected in this study was <500 μm, which poses a risk of being swallowed by organisms at the surface and bottom of the water (*Kasamesiri & Thaimuangphol, 2020*). In the dry season, smaller sizes (<500 μm) were found than in the wet season, which could be because of fragmentation by sunlight, wind, and waves in the dry season which break down larger pieces of fiber into smaller pieces due to the high temperatures of this season. *Kye et al. (2023)* also reported that aging and fragmented MPs become smaller. In addition, the smaller the MP, the larger the surface area that can absorb more contaminants (*Zhou, Liu & Wang, 2019*).

Color: the color of MP may indicate its source (*Chinfak et al., 2021*), for example white and blue fiber may result from the degradation of fishing gear, such as line and nets (*Oo et al., 2021*). Blue, black, and transparent color of fishing net are commonly used in Songkhla Lagoon. Other colors found in this study are likely from land base activities. The color of MPs may alter the possibility of being ingested by organisms (*Jovanović, 2017*). The colors of MPs found in catfish and shrimps caught in Songkhla Lagoon were black, blue, transparent, and red (*Pradit et al., 2021*).

## Polymer identification of MP

FTIR spectroscopy is the most commonly used instrument for polymer identification. *Cutroneo et al. (2020)* suggested that spectroscopic instruments can be used with a microscope to identify small dimensions (<1 mm). In this study, five polymer types were found. In the wet season, rayon and polymer were the most common while rayon and PET were the most common in the dry season. Interestingly, rayon was high in both seasons. Rayon and polyester are starting materials for textiles. Rayon fibers are lightweight and drift with the currents into the open sea, eventually entering the bodies of aquatic animals (*Azad et al., 2018*). Textile fibers were also discovered in the stomachs of fish in Songkhla Lagoon (*Pradit et al., 2021*). MPs from these synthetic clothes may be released due to chemical and mechanical stresses during washing processes in washing machines and then ultimately enter the river and the lagoon (*Choi et al., 2021*; *Fontana, Mossotti & Montarsolo, 2020*). However, PET and PP polymer types could come from fishing gear

or fishing nets since the lagoon itself is rich in fish species and fishing activities. PET has a larger density (1.38 g/cm³) than water, hence it tends to sink in water. However, in this study, we discovered it in the surface layer. A possible explanation is that the lagoon is quite shallow, and the hydrological characteristics and human activity (*e.g.*, fishing, transportation) can cause turbulence in the water, causing PET to diffuse and float to the surface layer and take some time to settle in the lagoon bottom. The density of PP (0.93 g/cm³) was lower than that of water, causing it to float in the surface water of the river and lagoon (*Pradit et al., 2023*; *Olarinmoye et al., 2020*; *Wakkaf et al., 2020*). Surprisingly, this study did not find polyamide (PA) or polyethylene (PE) in the surface water similar to the report in Malaysia estuary (*Zaki et al., 2021*) and polystyrene (PS) as reported in Lagoons of Northern Tunisia (*Wakkaf et al., 2020*) and in Southwest Nigeria (*Olarinmoye et al., 2020*).

In general, FTIR technique is useful to identify some chemical functional groups that would be active by infrared radiation like organic functional groups or metal oxides with low bonding energy. All common plastics, PE, PP, PET, PVC *etc.*, are active in IR technique, therefore, it widely used to identify and classify types of plastics. However, it is quite difficult to identify an unknown sample with complex components. In terms of plastic materials, characteristic peaks of each plastic are useful to confirm the plastic structure. For example, PE shows characteristic peaks of C-H vibration at 2,914, 2,846, 1,474 and 720 cm⁻¹. While PP shows the most similar absorption peak with PE, it has a strong peak at 1,378 cm⁻¹ to represent a vibration of −CH₃ to classify between these two polymers. PVC shows its characteristic peak at 1,245 cm⁻¹ referred to a vibration of C-Cl bond. FTIR pattern of PET is more complicated when compared to other during different chemical bond in the structure. Therefore, FTIR technique could be used to confirm chemical bonding in polymer samples due to their different characteristic peak. In the past three decades, the impact strength of PP has been improved by blending it with multiple elastomers such as ethylene–propylene random copolymer (EPR) (*Nitta et al., 2000*). Poly (ethylene-propylene) or copolymer between PE and PP is widely used or prepared to obtain desired properties. A good compatibility of the blends between PE and PP could be obtained due to a similarity of chemical structures. The copolymer between PE and PP is generally prepared to control amount of crystallinity which influences a different mechanical and thermal properties.

It was notable, that DEHP and DEG were observed in surface water in the wet season. DEHP is a plasticizer and DEG is a polyester-based polyester. Both DEHP and DEG can be found in several products such as furniture materials, personal care products, cosmetics, and medical devices. DEHP is an endocrine disruptor, an exogenous substance or a mixture that alters the functions of the endocrine system and consequently causes deleterious health effects on an intact organism, its progeny, or subpopulations (*Nohynek et al., 2013*). Since MPs were found in the water, they will likely eventually enter the food chain.

**Table 2 Comparison of MP abundance in different lagoons.**

| Location | Type area | Abundance | Polymer | Reference |
|----------|-----------|-----------|---------|-----------|
| Southern Thailand | River | 0.24 ± 0.11 to 0.41 ± 0.08 particles/L | PP, PET, nylon, PE, rayon, PDMS, copolymer | *Pradit et al. (2023)* |
| Southwest Nigeria | Lagoon | 139–303 particles/L | PP, PS, PE, polyester, polyacrylic | *Olarinmoye et al. (2020)* |
| South of Brazil | Lagoon | 0.0846 g/400L | PE, PFTE, LDPE, HDPE | *e Silva & de Sousa (2021)* |
| Northern Tunisia | Lagoon | 453.0 ± 335.2 items/m$^3$ | PE, PP, PET, CP, NL, PS, | *Wakkaf et al. (2020)* |
| Malaysia | Estuary | 0.5 to 4.5 particles/L | PA, PE | *Zaki et al. (2021)* |
| Malaysia | Seitu Wetland | 0.36 item/L | PP | *Ibrahim et al. (2021)* |
| China | Estuary | 23.1 ± 18.2 item/100 L | PE, PP, PVC, PA, ABS, PUR, PS, PC, SAN, ASA | *Xu et al. (2018)* |
| Southern Thailand | Lagoon | 0.43 ± 0.06 items/L in wet season, 0.34 ± 0.03 items/L in dry season | PET, copolymer, polyester, PP, rayon | This study |

## Water quality in surface water in relation to MP

Total dissolved solid (TDS), conductivity, and salinity levels were found to vary significantly between seasons. All followed the same trend, with a high value in the dry season and a low value in the wet season, despite the fact that the average surface water temperature does not fluctuate significantly. In the dry season, less rainfall in the summer leads to reduced runoff, while seawater intrusion to the lagoon results in high TDS. Moreover, the major source of the suspected matter in the lagoon is from runoff which causes high visibility (transparent water) in dry season.

It's noteworthy, that although though there is a high level of TDS, the water is still clear because the dissolved particles (smaller than 2 μm) are similar in size to nanoplastics (size varying between 1 nm and 1 μm). However total suspended solid (TSS) data, which would reveal the turbidity of the water and correlate with its water transparency, unfortunately was not collected. Nevertheless, Microorganisms are able to degrade TDS and their activities could be the cause of changes in TDS levels (*Adjovu, Stephen & Ahmad, 2023*). Furthermore, they can consume TDS during biological processes. Following the Thai standards for water (in the area for the conservation of natural life), the dissolved oxygen from this study was mostly within the criterion (>4 mg/l), except at the Samrong river mouth (station 7). It is noted that hypoxia, low or depleted oxygen in a water column, occurred in both seasons, in which aquatic organisms cannot be sustained. Even though the canal has a shorter length (about 4 km) than U-Taphao (about 120 km) and Phawong (about 5.5 km), the dense population in the area means that sewage tends to drain into the shallow canal (<100 cm). From visual observation, the water color is black and smells of hydrogen sulfide. However, the other sample site in the lagoon had a DO range of about 3–9 mg/l.

MP concentrations in both seasons were analyzed in relation to water quality parameters (temperature, pH, salinity, conductivity, transparency, TDS, and DO), as shown in Table 2 and Figs. 8 and 9. Conductivity has a strong correlation with TDS (wet season: r = 0.982; dry season: r = 1.000) and salinity (r = 0.988), while salinity has a strong

correction with TDS in the dry season (r = 0.998). Transparency has a high positive relationship with DO in the dry season (r = 0.833). It is noted that there is a negative correlation between MP and TDS during the wet season (−0.806, $p < 0.001$).

## CONCLUSIONS

The abundance, distribution, and characteristics of MP in the surface water together with the water quality of Songkhla Lagoon, Thailand, in the wet and dry seasons was investigated. The results show that MPs were found at all stations (mouth of the river, center of the lagoon, and the mangrove area). More MPs were found during the wet season than in the dry season. Our findings highlight that high MP concentrations in the mouth area of U-Taphao River (stations 1 and 10) indicate that it is a hot spot of MP in surface water of the lagoon followed by the mangrove site (station 5) in both seasons. Fiber was the most dominant shape. Blue and black color were the major MP colors found in surface water. From FTIR determination, five types of polymers were found consistent with PET, poly (ethylene:propylene), polyester, PP, and rayon. Additionally, DEHP and DEG were observed in surface water in the wet season. From the water quality result, we found hypoxia (low oxygen) in Samrong canal (station 7). According to this study, water quality was a significantly different between the rainy and dry seasons ($p < 0.05$). This report can raise awareness of the situation of MPs in the surface water of the lagoon and highlights the need for regular monitoring. Furthermore, it is necessary to characterize the influence of the river contamination on MP concentrations in the lagoon.

### Funding
This work was financially supported by the National Science, Research, and Innovation Fund (NSRF) and Prince of Songkla University (Grant No. ENV6601203S). The funders had no role in study design, data collection and analysis, decision to publish, or preparation of the manuscript.

### Grant Disclosures
The following grant information was disclosed by the authors:
National Science, Research, and Innovation Fund (NSRF).
Prince of Songkla University: ENV6601203S.

### Competing Interests
The authors declare that they have no competing interests.

### Author Contributions
- Siriporn Pradit conceived and designed the experiments, performed the experiments, analyzed the data, authored or reviewed drafts of the article, and approved the final draft.
- Prakrit Noppradit conceived and designed the experiments, authored or reviewed drafts of the article, and approved the final draft.

- Kittiwara Sornplang performed the experiments, analyzed the data, prepared figures and/or tables, and approved the final draft.
- Preyanuch Jitkaew performed the experiments, analyzed the data, prepared figures and/or tables, and approved the final draft.
- Thanakorn Jiwarungrueangkul analyzed the data, authored or reviewed drafts of the article, and approved the final draft.
- Dudsadee Muenhor conceived and designed the experiments, authored or reviewed drafts of the article, and approved the final draft.

## Data Availability

The raw data are available in the Supplemental Files.

## Supplemental Information

Supplemental information for this article can be found online at http://dx.doi.org/10.7717/peerj.17822#supplemental-information.

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
