# Peer review of "Occurrence and abundance of microplastics in surface water of Songkhla Lagoon"

_PeerJ, doi:10.7717/peerj.17822_

## Round 0.1 · original submission · Major Revisions

I believe that your manuscript is worthy of publication. It does require major revision, however. Both reviewers had significant comments on the methodology, analysis, and, or the results - these problems need to be addressed.

Reviewer 1 has requested that you cite specific references. You may add them if you believe they are especially relevant. However, I do not expect you to include these citations, and if you do not include them, this will not influence my decision.

Reviewer 1 ·

Basic reporting

The manuscript entitled "Occurrence and abundance of microplastics in surface water of Songkhla Lagoon" is a welcome addition to the growing literature on microplastics pollution especially in lagoons, wetland or estuary area. However here is my observation and suggestions going through the manuscript:

Gaps and scope of the study:

1. Line 27-28: authors stated that MPs as global environmental concern, so how to relate this statement with the current manuscript?

2. Similarly, the abstract needs to be improved (line 28-30)- what do you meant by comprehensive contamination lever? it is comprehensive because of correlation with physical water parameters? or because of spatio-temporal? or because of large volume of sample/ representative sample?

3. The abstract should be written in a flow/sequence from general introduction, importance of the study, nature and methods of study, major results, important conclusions and impacts/implications.

4. The choice of the keywords need to be revisited: microplastics and waters shouldn't be in the keywords. Find more accurate keywords based on the manuscript.

5. Line 94-96: not clear what authors meant by this "ere is growing unequivocal scientific evidence
95 indicating that marine creature..." and how does this applies to this current study?

6. It is suggested that authors add and discuss more on the gaps, especially on the lagoons, wetlands point of view, and broader context of Asia/Southeast Asia (Table 1). The discussion on seems to be on the surface level. How does the findings add on to the very much baseline study in the region? What is the novelty of the study, especially for international or regional point of view? Additionally some of major references in the similar set up has not being compared and discussed in the Table 1.(https://doi.org/10.1016/j.marpolbul.2018.06.020, https://doi.org/10.1039/c9em00148d,
https://doi.org/10.1016/j.scitotenv.2021.147809)

Discuss why this current manuscript present lower amount /higher amount compare to the previous study? is there any environmental/geological factors related to this?

Experimental design

1. It is not clear HOW authors carried out the sampling/sample collection for all 20 surface water sample (Line 147)- This is critical - authors mentioned using bucket sampling of 50L and (assuming it was then pre concentrated by net mesh) to how many L? and how this 50L enough sample size for the representative sample?
2. Is there any control for field and laboratory analysis? Field blank/ procedural blank/ negative or positive control?
3. Sample extraction: how much peroxide reagent being used? and the concentration?
4. How many samples underwent polymer analysis by FTIR? what is the analysis method for FTIR used?
What is the detector used? and the limitation of size? and library of IR used? is this micro-FTIR or only FTIR? if only FTIR how authors examine the smaller size fraction of MPs?

Validity of the findings

1. Copolymer is usually the additives used in the polymer industry. does this suggest the coating of the samples could be the reason? since FTIR supposed to analyse the polymer backbone, but if polymer was really stained with other additional polymeric substances, it will overshadowed the backbone of the polymer. Please add some related discussion on this, since this could help other researchers too if they have similar findings,.

Reviewer 2 ·

Basic reporting

Introduction
In general, this section is not well organized. The background is not clear an ambiguous.
The topic sentence and the main idea of the paragraph are not stated systematically. One paragraph contains several main ideas that are not connected at all. One sentence said something, and the next sentence said another topic that was not connected with the previous sentence.
Systematically, the delivery of the flow of thought between paragraphs is also still lacking.
Some sentences are still grammatically incorrect, and some sentences are too long.

Experimental design

Methods need to be described in detail, along with information on sampling replication. Sampling methods may not give a more accurate estimation or a good representation.

Validity of the findings

Figures are relevant, high quality, well labelled & described.
Results: The data looks robust. Although the methodology has a slight flaw, this study still contributes to new information.

Discussion: The author does not explain the significant findings or the relevance of the results. It does not show a rational argument based on basic knowledge in support of the overall conclusion.

Conclusions are not appropriately stated, and do not reflect the research title, and do not connect to research question.

Additional comments

In general, the manuscript is not well written, especially considering the background of the study and the discussion section. However, the results presented well and showed significant findings.
TThe manuscript needs to be reconstructed, and the discussion section should show in-depth arguments supporting the significance and relevance of the findings.
Rewrite the conclusion based on the results, not otherwise.

Annotated reviews are not available for download in order to protect the identity of reviewers who chose to remain anonymous.

---

## Round 0.2 · Minor Revisions

Please add the text (or something similar) that the reviewer suggested regarding experimental design

Reviewer 1 ·

Basic reporting

The authors did very well in addressing all comments from both reviewers, this current version conveys that the necessary improvements have been made and that the manuscript is ready for publication after one more additional amendment on conclusion/end of discussion. Please add the limitation of the study and way forward in microplastics research in lagoon/tropical estuary.

Experimental design

In the response of our previous comments: "Is there any control for field and laboratory analysis? Field blank/ procedural blank/ negative or positive control?"
Author’s response: Thank you for your valuable comment. We understand that all of the controls you mentioned are important., Unfortunately, we do not perform field blanks, procedural blanks, negative or positive controls. However, we prevent contamination in both laboratory and field activities.

Please add this information (in revised form) to the conclusion or somewhere in the discussion- please also add other limitations of the study that need to be addressed, so this will create a new gap of knowledge for other researchers, and especially when relate to possibility of overestimation/underestimation of using current technique.

Validity of the findings

OK

---

## Round 0.3 · accepted · Accept

Thank you for responding to all of the suggested reviewers comments. I agree with the last reviewer that the lack of field controls/blank data leaves open the need to refine this science in the future.